# Enhancing User Intent Capture in Session-Based Recommendation with Attribute Patterns

**Xin Liu**[1][*] **Zheng Li**[2] **Yifan Gao**[2] **Jingfeng Yang**[2]
**Tianyu Cao**[2] **Zhengyang Wang**[2] **Bing Yin**[2] **Yangqiu Song**[1][†]
[1]Department of Computer Science and Engineering, HKUST    [2]Amazon.com Inc
`{xliucr,yqsong}@cse.ust.hk`
`{amzzhe,yifangao,jingfe,caoty,zhengywa,alexbyin}@amazon.com`

## Abstract

The goal of session-based recommendation in E-commerce is to predict the next item that an anonymous user will purchase based on the browsing and purchase history. However, constructing global or local transition graphs to supplement session data can lead to noisy correlations and user intent vanishing. In this work, we propose the Frequent Attribute Pattern Augmented Transformer (FAPAT) that characterizes user intents by building attribute transition graphs and matching attribute patterns. Specifically, the frequent and compact attribute patterns are served as memory to augment session representations, followed by a gate and a transformer block to fuse the whole session information. Through extensive experiments on two public benchmarks and 100 million industrial data in three domains, we demonstrate that FAPAT consistently outperforms state-of-the-art methods by an average of 4.5% across various evaluation metrics (Hits, NDCG, MRR). Besides evaluating the next-item prediction, we estimate the models' capabilities to capture user intents via predicting items' attributes and period-item recommendations.

## 1 Introduction

With the explosive demand for E-commerce services [44, 12, 23, 11, 43], numerous user behaviors are emerging. Understanding these historical action records is critical in comprehending users' interests and intent evolution, particularly in a cold-start regime that lacks sufficient context. This has spurred research on session-based recommendations (SBR) [8, 35, 37, 13] that capture user-side dynamics from a short-time period (namely a session) using temporally historical information. Numerous SBR algorithms have been proposed, ranging from sequence-based methods [27, 8, 16, 19, 31, 28, 9] to graph-based methods [35, 38, 34, 37, 17, 30] for learning dynamic user characterization. However, both lines of methods have their limitations. Specifically, sequence-based methods treat each user behaviors in a session as an action sequence and model the local dependencies inside. This can only capture users' preference evolution via chronological order while failing to identify the complex non-adjacent item correlation, especially when the session length is insufficient to support temporal prediction [33]. To address this issue, graph-based methods adopt a higher perspective by introducing a global item transition graph, which aggregates local session graphs constructed from historical session sequences. Thus, a newly-emerging short session sequence can benefit from the global topology (e.g., global co-occurrences) and representations (e.g., item semantics) [34]. Unfortunately, existing session graph construction ignores temporal signals. Figure 1 shows that two different sessions result in the same session graph, leading to vanishing of user intent variation during sequence-to-graph conversion. And such global graphs are fragile due to noise from random clicks.

---

[*]Work was done during Xin's internship at Amazon. Corresponding author: Xin Liu and Zheng Li.
[†]Prof. Yangqiu Song is a visiting academic scholar at Amazon.

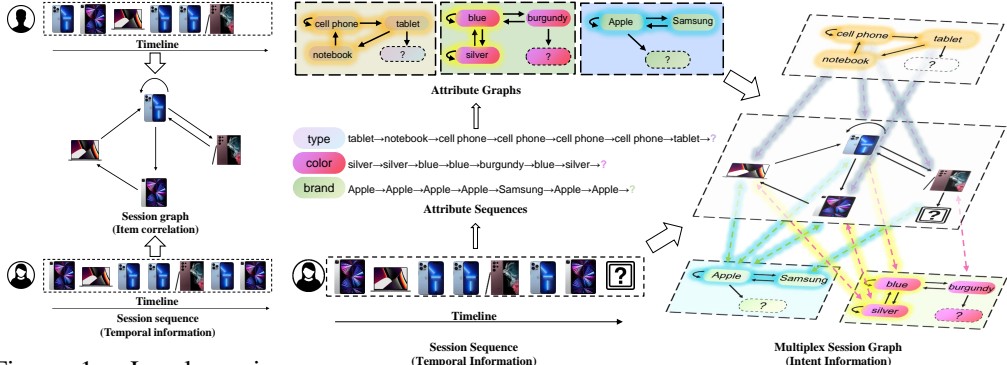

Figure 1: Local session graph construction.

Figure 2: A session graph enriched by multiplex attribute graphs.

Besides the global item transition graph, there are other possible solutions for item correlations. One feasible solution involves item-side knowledge. Items can be connected through shared attributes, such as being manufactured by the same company [28]. We argue that the current use of item-side metadata provides little assistance in SBR models as user intent may change over time. However, such meta-data are still useful from the view of graphs. As illustrated in Figure 2, session attributes can be organized into attribute graphs and anchored to the local session graph to create multiplexes. Such multiplexes provide several intent clues, in addition to item-side correlations. For example, the color pattern *silver ↔ silver ↔ blue ↔ blue* reveals the user's color intent, while the brand pattern *Apple ↔ Apple ↔ Samsung* implies a potential change in intent. Different sessions can benefit from shared attribute topologies and representations, which can ultimately entail implicit high-order correlations. However, the session graph is essentially a general conditional random field, and optimization becomes intractable due to the large candidate size. Graph neural networks (GNNs) also face challenges due to the possibility of over-smoothing and data noise [46, 18, 45, 32].

To alleviate the aforementioned issues, we propose a novel framework called Frequent Attribute Pattern Augmented Transformer (**FAPAT**) that considers highly frequent attribute patterns as supplementary instead of directly learning multiplex graphs. In traditional graph learning, graphlets (such as triangles, triangular pyramids, etc.) have been proven to be useful features in graph classification and representation learning [29, 15]. Therefore, we employ frequent graph pattern mining algorithms to find consequential graphlets and view them as compact hyper-edges (e.g., *cell phone*, *tablet*, *notebook* in Figure 2). We then use these attribute patterns as accessible memory to augment session sequence encoding. Before encoding patterns, we use Jaccard similarities to rank and retrieve the most highly correlated patterns, which significantly reduces the graph density and computational cost. It has been shown that GNNs can estimate the isomorphism and frequency of substructures [39, 21, 20]. Thus, we leverage multi-head graph attention to learn pattern and local session graph representations in the aligned space [10, 22]. To incorporate temporal signals and capture user intents, we distribute graph representations back to session sequences and use external pattern memory to augment sequence representations via memory attention with relative position bias. Finally, the sequence is fully fused by a transformer block. In other words, graph information is used to aggregate attribute patterns, while temporal actions are used to encode items.

To validate the effectiveness of FAPAT, we conduct extensive experiments on two public benchmark datasets and three real-world large-scale industrial datasets with around 100 million clicks, and experimental results demonstrate significant improvement with an average boost of 4.5% across various evaluation metrics (Hits, NDCG, MRR). Compared with baselines, the attribute pattern density can significantly relieve over-smoothing. Besides, we also extend evaluation to attribute estimations and sequential recommendations to measure the model capability to capture user intents. Code and data are availiable at `https://github.com/HKUST-KnowComp/FAPAT`.

## 2 Related Work

***Neural Methods for Session-based Recommendation Systems***. Table 1 presents a summary of the distinctions between current neural techniques and our novel FAPAT. The concept of *Temporal Information* implies Markov decision processes with previous histories. The technique of *History Attention* employs attention for learning long-distance sequences. The approach of *Local Session*

Table 1: Comparison with existing popular methods.

| Methods | Temporal Information | History Attention | Local Session Topology | Global Item Correlation | Attribute Association |
|---|---|---|---|---|---|
| FPMC | ✓ | ✗ | ✗ | ✓ | ✗ |
| GRU4Rec | ✓ | ✗ | ✗ | ✗ | ✗ |
| NARM | ✓ | ✓ | ✗ | ✗ | ✗ |
| STAMP | ✓ | ✓ | ✗ | ✗ | ✗ |
| CSRM | ✓ | ✓ | ✗ | ✓ | ✗ |
| S3-Rec | ✓ | ✓ | ✗ | ✓ | ✓ |
| M2TRec | ✓ | ✓ | ✗ | ✓ | ✓ |
| SR-GNN | ✗ | ✗ | ✓ | ✗ | ✗ |
| GC-SAN | ✗ | ✓ | ✓ | ✗ | ✗ |
| S2-DHCN | ✗ | ✗ | ✓ | ✓ | ✗ |
| GCE-GNN | ✗ | ✗ | ✓ | ✓ | ✗ |
| LESSR | ✓ | ✗ | ✓ | ✗ | ✗ |
| MSGIFSR | ✓ | ✓ | ✓ | ✗ | ✗ |
| FAPAT | ✓ | ✓ | ✓ | ✓ | ✓ |

*Topology* involves modeling session sequences from the view of session graphs. Lastly, *Global Item Correlation* and *Attribute Association* place emphasis on capturing item-side and attribute knowledge.

*Sequence-based Models.* FPMC [27] uses first-order Markov chain and matrix factorization to identify sequential patterns of long-term dependencies. However, the Markov-based method usually has difficulty exploring complicated temporal patterns beyond first-order relationships. Recently, neural networks have shown power in exploiting sequential data in SBR tasks, such as GRU4Rec [8]. NARM [16] extends GRUs with attention to emphasize the user's primary purchase purpose. Similarly, STAMP [19] uses an attention-based memory network to capture the user's current interest. These attention-based models separately deal with the user's last behaviors and the whole session history to detect the general and latest interests. But they mainly focus on the user's preference from a temporal view but ignore the item correlations. Pre-training techniques [47] and multi-task learning [28] also demonstrate effectiveness in injecting item metadata to embeddings and predicting item attributes. Besides, some recent sequence-based approaches leverage generative pretrained language models to provide explicit explanations for recommendation systems [4, 6].

*Graph-based Models.* Graph neural networks (GNNs) have recently been explored in SBRs due to the substantial implications behind natural transition topologies. SR-GNN [35] adopts a gate GNN to obtain item embeddings over the local session graph and predict the next item with weighted sum pooling, showing impressive results on benchmark data. Some advanced variants have further boosted performance, such as GC-SAN [38] with self-attention mechanism and FGNN [24] with weighted attention graph layers. To acquire further collaborative information, S2-DHCN [31] constructs line graphs to capture correlations among neighbor sessions, and GCE-GNN [34] directly applies a graph convolution over the global transitions to aggregate more relevant items for local sessions. However, GNN-based methods still face challenges in capturing temporal signals, filtering noise, and leveraging implicit high-order collaborative information. Other methods, such as LESSR [1] and MSGIFSR [7], have also shown significant improvement by building multigraphs and shortcut graphs for session representation learning and user intents from different granularities, respectively.

***Pattern Mining for Recommendation Systems***. Pattern mining is an important data mining technique with board applications. In recommendation systems, sequential pattern mining assists in analyzing customer purchase behaviors through frequent sequential patterns. Such mining focuses on item patterns with frequencies above a threshold in all sessions, which reduces the diversity of the recommended items [2]. Personalized sequential pattern mining [42] effectively learns user-specific sequence importance knowledge and improves the accuracy of recommendations for target users. It can be challenging to generalize to SBR systems when neural networks have already implicitly captured such behavior patterns. But attribute graph patterns still need to be explored in SBR.

# 3 Background and Motivations

## 3.1 Problem Definition

Session-Based Recommendation (SBR) assumes that users' historical behaviors outside the current session are inaccessible, in contrast to general recommendations. For example, users do not log in due to user privacy and security reasons. SBR predicts the next item that an anonymous user is most

likely to click on or purchase based on historical behaviors within a short period. Despite the lack of personal profiles, this universal setting can better reflect the quality of item-side recommendations. Suppose that there are $N$ unique items in the database, and each session is represented as a repeatable sequence of items $S = [v_1, v_2, \ldots, v_L]$, $v_i \in \mathcal{V}$ $(1 \leq i \leq L)$ represents the $i$-th behavioral item of the anonymous user within session $S$, where $\mathcal{V}$ is the item set collected from overall sessions, and $L$ is the length of the session. Given a session $S$, the goal is to recommend the top-$K$ items $(1 \leq K \leq N)$ that have the highest probabilities of being clicked by the anonymous user.

## 3.2 Session Graphs and Transition Graphs

Session sequence modeling is not always sufficient for SBR as it only reflects transitions from the user side. To account for item correlations, SR-GNN [35] converts session sequences to session graphs. Each session graph $\mathcal{G}_S = (\mathcal{V}_S, \mathcal{E}_S)$ is a directed graph with node set $\mathcal{V}_S \subseteq \mathcal{V}$, consisting of unique items in the session, and edge set $\mathcal{E}_S$, recording adjacent relations between two items in session $S$. Edge weights can be normalized by indegrees or outdegrees to model transition probabilities. GCE-GNN [34] extends this graph modeling by merging all session graphs as a global transition graph, which aggregates more item correlations but faces over-smoothing and data noise [46, 18].

Instead of modeling global item transitions, we enrich session graphs with attributes and patterns. Assume there are $M$ different kinds of attributes, and the $m$-th attribute type $\mathcal{A}^{(m)}$ $(1 \leq m \leq M)$ consists of $|\mathcal{A}^{(m)}|$ possible values. Thus, each item $v \in \mathcal{V}_S$ has attribute list $[a_v^{(1)}, a_v^{(2)}, \cdots, a_v^{(M)}]$, where $a_v^{(m)} \in \mathcal{A}^{(m)}$ denotes the $m$-th attribute value of $v$. Each session sequence $S$ corresponds to $M$ attribute histories, with the $m$-th attribute sequence denoted as $S^{(m)} = [a_{v_1}^{(m)}, a_{v_2}^{(m)}, \cdots, a_{v_L}^{(m)}]$. Using the sequence-to-graph transform, we convert $S^{(m)}$ to $\mathcal{G}_S^{(m)}$, whi is usually denser than the session graph $\mathcal{G}_S$. Finally, the $M$ attribute sequences are separately transformed into $M$ attribute session graphs in different property-specific channels, anchored in items $\mathcal{V}_S$. Finally, we add edges for item $v_i$ and attribute $a_{v_i}^{(m)}$ to construct a multiplex session graph, preserving attribute values and transitions. We represent the session graph with attributes as the multiplex $\mathcal{G}_S^{\mathcal{A}}$ (as Figure 2).

## 3.3 Frequent Pattern Mining

Frequent pattern mining aims to extract inductive clues from data to comprehend data distributions, which includes two main categories: sequential pattern mining and graph pattern mining. The former is concerned with sequence databases composed of ordered elements, while the latter statistics the important graph structures. For two sequences $S' = [v'_1, v'_2, \cdots, v'_{L'}]$ and $S = [v_1, v_2, \cdots, v_L]$, we refer to $S'$ as the pattern of $S$ if $S'$ is a subsequence of $S$. Similarly, a graph $\mathcal{G}_{S'} = (\mathcal{V}_{S'}, \mathcal{E}_{S'})$ is a subgraph of $\mathcal{G}_S = (\mathcal{V}_S, \mathcal{E}_S)$ if $\mathcal{V}_{S'} \subseteq \mathcal{V}_S$ and $\mathcal{E}_{S'} \subseteq \mathcal{E}_S$. Compared with sequence pattern mining, graph pattern mining is more general since it involves the structural topology and attribute information. For instance, as depicted in Figure 2, {*cell phone*, *tablet*, *notebook*} corresponds to a triangle. It may be challenging to discover the triangle from the temporal sequence, but it is a vital clue from the graph view. Thus, we stick on frequent graph patterns rather than sequence patterns.

# 4 Methodology

We present the Frequent Attribute Pattern Augmented Transformer (FAPAT), a novel framework that captures user intents and item correlations. Our method is built upon session sequences and corresponding attribute graphs. Initially, we mine frequent attribute patterns from the attribute graphs to explore coarse-grained item correlations. These patterns are then used as memory to enhance the session encoder, which consists of graph-nested transformer layers. Figure 3 illustrates the overview.

## 4.1 Frequent Attribute Pattern Acquisition

In this subsection, we describe how to extract frequent patterns from training recommendation sessions. The aim of frequent pattern mining is to minimize the impact of random clicks in the global transition graph and avoid over-smoothing when learning multiplex attribute graphs. To achieve this, we design a mining-filtering paradigm to ensure representativeness.

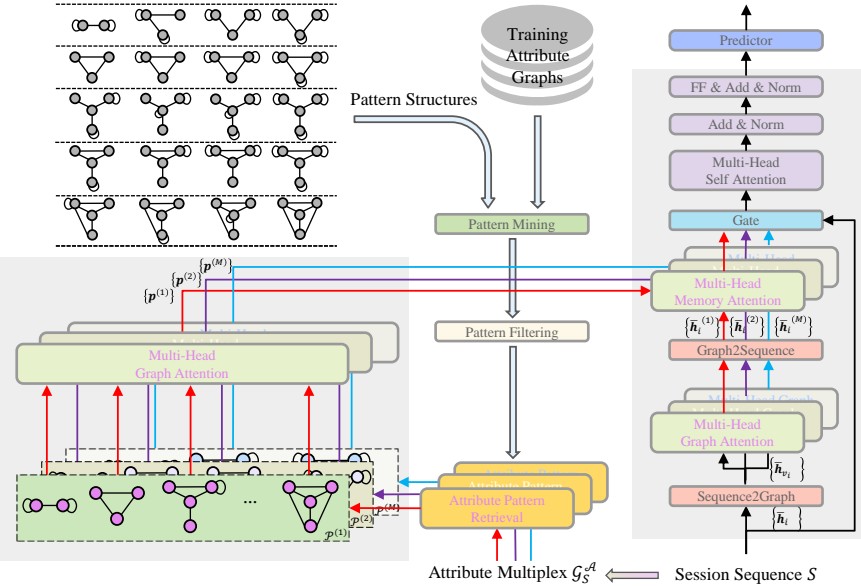

Figure 3: An overview of FAPAT.

### 4.1.1 Graph Pattern Mining

Small graph patterns, also called motifs or graphlets, are valuable features in graph learning and property prediction [29, 15], exhibiting strong statistical correlations between graph structures and node semantics. In this study, we focus on similar graphlet structures but extend to SBR scenarios. We collect patterns consisting of no more than four nodes (representing different attribute values), and further restrict them to those containing either a circle or a triangle to significantly reduce the number of candidates. We adopt gSpan [40] to acquire undirected patterns from attribute session graphs and keep patterns belonging to one of twenty types shown in Figure 3.

### 4.1.2 Loose Pattern Filtering

However, complex patterns may contain smaller ones. For instance, the first pattern in Figure 3 is a subgraph of the second. While each subgraph has an equal or higher frequency than its supergraph, such loose patterns do not convey much information. To eliminate them, we employ VF2 [3] subgraph isomorphism algorithm to filter. If a pattern $P'$ is a subgraph of another pattern $P$, then $P'$ is excluded from the pattern candidates. Finally, we retain compact and frequent patterns for session encoding.

### 4.2 Intent-aware Sequence Encoding

To recommend items of high interest to users, we use a GAT-based encoder to learn pattern representations from the item side, which are then served as memory to augment session encoding.

### 4.2.1 Relevant Graph Pattern Retrieval

The input item sequence is converted to a multiplex session graph representing the transitional information of different item attributes, as depicted in Figure 1. To improve graph representations by utilizing frequent attribute subgraphs, we retrieve relevant patterns from those mined in §4.1. For the $m$-th attribute type $\mathcal{A}^{(m)}$ ($1 \leq m \leq M$), we denote an arbitrary subgraph mined from the previous step as $\mathcal{G}_P^{(m)} = (\mathcal{V}_P^{(m)}, \mathcal{E}_P^{(m)})$. Then we retrieve at most $I$ subgraph patterns that have the most considerable Jaccard similarities to the transition graph $\mathcal{G}_S^{(m)}$ as Eq. (1). This can be done within $\mathcal{O}(|\mathcal{V}_S^{(m)}| \times |\mathcal{V}_P^{(m)}|)$, but this is always linear because of $|\mathcal{V}_P^{(m)}| \leq 4$.

$$\text{Jaccard}\big(\mathcal{V}_S^{(m)}, \mathcal{V}_P^{(m)}\big) = \frac{|\mathcal{V}_S^{(m)} \cap \mathcal{V}_P^{(m)}|}{|\mathcal{V}_S^{(m)}| + |\mathcal{V}_P^{(m)}| - |\mathcal{V}_S^{(m)} \cap \mathcal{V}_P^{(m)}|}. \tag{1}$$

#### 4.2.2 Attribute Pattern Representation

After retrieving the relevant patterns for the corresponding multiplex session graph, we encode them using multi-head relational graph attention for further memory augmentation. For a pattern $\mathcal{G}_P^{(m)}$ for the $m$-th attribute, we compute the attention weight for two arbitrary nodes $a_i^{(m)}$ and $a_j^{(m)}$ by:

$$\alpha_{ij}^{(m)} = \mathrm{softmax}\Big(\frac{\mathrm{LeakyReLU}\big(\boldsymbol{r}_{ij}^{(m)^\top}\big(\boldsymbol{e}_{a_j^{(m)}} \circ \boldsymbol{e}_{a_i^{(m)}}\big)\big)}{\sum_{a_k^{(m)} \in \mathcal{N}(a_i^{(m)})} \mathrm{LeakyReLU}\big(\boldsymbol{r}_{ik}^{(m)^\top}\big(\boldsymbol{e}_{a_k^{(m)}} \circ \boldsymbol{e}_{a_i^{(m)}}\big)\big)}\Big), \tag{2}$$

where $\boldsymbol{e}_{a_i^{(m)}}$ denotes the embedding of $a_i^{(m)}$, $\circ$ indicates element-wise multiplication, $\mathcal{N}(a_i^{(m)})$ represents the neighbors of $a_i^{(m)}$ (including itself), and $\boldsymbol{r}_{ij}^{(m)}$ corresponds to the relation-specific vector. Then the representation of $\overline{\boldsymbol{h}}_{a_i^{(m)}}$ for node $a_i^{(m)}$ in the pattern is aggregated by:

$$\overline{\boldsymbol{h}}_{a_i^{(m)}} = \sum_{a_j^{(m)} \in \mathcal{N}(a_i^{(m)})} \alpha_{ij}^{(m)} \boldsymbol{e}_{a_i^{(m)}}. \tag{3}$$

After computing the representation of each node in pattern subgraph $\mathcal{G}_P^{(m)}$, a pooling layer (e.g, average pooling) over all node representations is to aggregate the pattern presentation for $\mathcal{G}_P^{(m)}$:

$$\boldsymbol{p}^{(m)} = \mathrm{Pool}\big(\{\overline{\boldsymbol{h}}_{a_i^{(m)}} | v_i \in \mathcal{G}_P^{(m)}\}\big) \tag{4}$$

#### 4.2.3 Attribute Memory Augmentation

Meanwhile, we also employ graph attention to obtain the native graph representations of the multiplex session graph $\mathcal{G}_S^{(m)}$. We apply this to each $m$-th attribute transition graph $\mathcal{G}_S^{(m)}$, computing node representations $a_i^{(m)}$ using Eq. (2) and Eq. (3) and denote the resulting node representation as $\overline{\boldsymbol{h}}_{v_i}^{(m)}$. To compute the aggregated attribute representation of each node $v_i$ in the original session graph $\mathcal{G}_S$, we combine its different representations from all attribute transition graphs by:

$$\overline{\boldsymbol{h}}_{v_i} = \frac{1}{M} \sum_m \overline{\boldsymbol{h}}_{v_i}^{(m)} + \boldsymbol{e}_{v_i}, \tag{5}$$

where $\boldsymbol{e}_{v_i}$ is the representation of $v_i$ by the item embedding lookup.

Suppose all retrieved attribute patterns associated with $\mathcal{A}^{(m)}$ are $\mathcal{P}^{(m)}$, whose representations from Eq. (4) are $\{\boldsymbol{p}^{(m)} | p^{(m)} \in \mathcal{P}^{(m)}\}$. To preserve temporal information when utilizing these patterns, we map graph representations $\{\overline{\boldsymbol{h}}_{v_i}^{(m)} | v_i \in \mathcal{V}_S\}$ to sequence representations $\{\overline{\boldsymbol{h}}_i^{(m)} | 1 \leq i \leq L\}$ with length $L$. Next, we concatenate all pattern representations $\{\boldsymbol{p}^{(m)} | p^{(m)} \in \mathcal{P}^{(m)}\}$, all session representations $\{\overline{\boldsymbol{h}}_i^{(m)} | 1 \leq i \leq L\}$, and two special embeddings $\boldsymbol{e}_{\mathrm{CLS}}$ and $\boldsymbol{e}_{\mathrm{MASK}}$ to separate the two parts and indicate the item to predict:

$$\left[\Big[\underset{p^{(m)} \in \mathcal{P}^{(m)}}{\|} \boldsymbol{p}^{(m)}\Big] \| \boldsymbol{e}_{\mathrm{CLS}} \| \Big[\underset{1 \leq i \leq L}{\|} \overline{\boldsymbol{h}}_i^{(m)}\Big] \| \boldsymbol{e}_{\mathrm{MASK}}\right].$$

Then, we augment sequence encoding with memory, taking inspiration from TransformerXL [5] and Memorizing Transformer [36]. As shown in Figure 4, we introduce T5 relative position bias [25] to distinct short and long-term histories. Furthermore, we utilize unidirectional attention to focus solely on the user's current intent instead of global intent, which may become increasingly noisy over time.

#### 4.2.4 User Intent Aggregation

To aggregate memory-augmented representations from $M$ different attributes and pattern sets, we employ gating mechanism:

$$\hat{\boldsymbol{h}}_i = \sum_{m=1}^{M} \beta_i^{(m)} \hat{\boldsymbol{h}}_i^{(m)} + \overline{\boldsymbol{h}}_i, \text{s.t. } \beta_i^{(m)} = \mathrm{softmax}(\boldsymbol{W}_\beta^{(m)} \overline{\boldsymbol{h}}_i^{(m)} + \boldsymbol{b}_\beta^{(m)}), \tag{6}$$

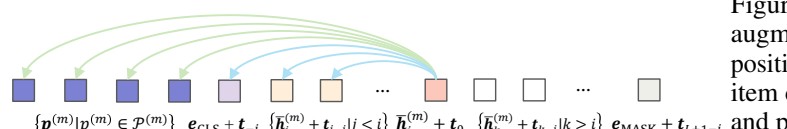

$\{\boldsymbol{p}^{(m)}|p^{(m)}\in\mathcal{P}^{(m)}\}\quad \boldsymbol{e}_{\text{CLS}}+\boldsymbol{t}_{-i}\quad \{\overline{\boldsymbol{h}}_j^{(m)}+\boldsymbol{t}_{j-i}|j<i\}\quad \overline{\boldsymbol{h}}_i^{(m)}+\boldsymbol{t}_0\quad \{\overline{\boldsymbol{h}}_k^{(m)}+\boldsymbol{t}_{k-i}|k>i\}\quad \boldsymbol{e}_{\text{MASK}}+\boldsymbol{t}_{L+1-i}$

Figure 4: Schema of pattern augmented attention: relative position bias is added, and one item can only access memory and previous histories.

where $\boldsymbol{W}_\beta^{(m)}\in\mathbb{R}^{d\times d}$ and $\boldsymbol{b}_\beta^{(m)}\in\mathbb{R}^d$ are trainable parameters ($d$ is the dimension of hidden states), $\hat{\boldsymbol{h}}_i^{(m)}$ is the memory-augmented results, and $\beta_i^{(m)}$ controls the importance of attribute patterns. While $\hat{\boldsymbol{h}}_i$ combines attribute information, it disregards long-range histories as $\overline{\boldsymbol{h}}_i^{(m)}$ solely accesses one-hop neighbors. Thus, we incorporate a transformer block to aggregate global session information:

$$\boldsymbol{H} = \text{Transformer}\Big(\Big[\hat{\boldsymbol{h}}_{\text{CLS}}\Big\|\Big[\underset{1\leq i\leq L}{\|}\hat{\boldsymbol{h}}_i\Big]\Big\|\hat{\boldsymbol{h}}_{\text{MASK}}\Big]\Big), \tag{7}$$

where $\hat{\boldsymbol{h}}_{\text{CLS}}$ and $\hat{\boldsymbol{h}}_{\text{MASK}}$ are directly taken from Eq. (6) since they do not belong to the original session but still observe the memory patterns and relative temporal signals.

### 4.3 Next-item Recommendation

Once the sequence representations are obtained, the next step is to predict the next item that may interest the user for clicking or purchasing. We adopt the approach used in previous work [35, 34, 37] to concatenate additional reversed positional embeddings as follows:

$$\boldsymbol{z}_i = \tanh\big(\boldsymbol{W}_z\big[\boldsymbol{h}_i\,\|\,\boldsymbol{t}_{L-i+1}\big]+\boldsymbol{b}_z\big), 0\leq i\leq L+1, \tag{8}$$

where $\boldsymbol{h}_i$ is the $i$-th item representation $\boldsymbol{H}[i]$ from Eq. (7), and $\boldsymbol{W}_z\in\mathbb{R}^{2d\times d}$ and $\boldsymbol{b}_z\in\mathbb{R}^d$ are trainable parameters. This concatenation of positional embeddings is intended to prioritize nearest intents over long-distance historical purposes. Figure 2 demonstrates this idea through two cases: the male user is more likely to be interested in Android cell phones since his last click was in that category, while the female user sticks to Apple products as she reviews iPhone and iPad once again.

We utilize the representation of the MASK to compute soft attention and then represent the session and the user's latest intent through a weighted average:

$$\boldsymbol{u} = \sum_{i=1}^{L}\gamma_i\boldsymbol{h}_i, \text{s.t. } \gamma_i = \boldsymbol{r}_\gamma^\top\text{sigmoid}(\boldsymbol{W}_\gamma\boldsymbol{z}_i + \boldsymbol{z}_{\text{mask}} + \boldsymbol{b}_\gamma), \tag{9}$$

where $\boldsymbol{W}_\gamma\in\mathbb{R}^{d\times d}$ and $\boldsymbol{b}_\gamma\in\mathbb{R}^d$ are trainable parameters. We compute the prediction of the next item using a similarity-based approach rather than a linear layer. This approach is similar to the optimization of Bayesian Personalized Ranking (BPR) [26]. Alternatively, we can train the model using cross-entropy minimization. Finally, the prediction is obtained by $\hat{y} = \text{argmax}_v(\boldsymbol{u}^\top\boldsymbol{e}_v)$.

## 5 Experiment

### 5.1 Setup

*Datasets*. We first evaluate our method on public benchmarks. *diginetica* contains the browser logs and anonymized transactions, *Tmall* collects anonymous users' shopping logs on the Tmall online website. We also acquire sessions from the browse and purchase logs from our E-commerce platform. We target at *beauty*, *books*, and *electronics* and gather 20-minute interactions within last successful purchases into one session after removing long-tail items. Appendix C provides more details.

*Baselines*. We compare our method with seven sequence-based baselines (FPMC [27], GRU4Rec [8], NARM [16], STAMP [19], CSRM [31], S3-Rec [47], and M2TRec [28]) and six graph-based baselines (SR-GNN [35], GC-SAN [38], S2-DHCN [37], GCE-GNN [34], LESSR [1], and MSGIFSR [7]). Model details are given in Appendix D. Each model is aligned with the official code implementation.

*Evaluation*. We evaluate SBRs as a ranking problem and employ Hits@$K$, NDCG@$K$, MRR@$K$ as standard metrics. Hits@$K$ measures the percentage of ranks up to and including $K$, while NDCG@$K$

Table 2: Performance evaluation for next-item prediction, where standard deviations are enclosed in brackets. The best and second-best results are respectively highlighted in bold and underlined. Methods that use attributes are marked with ‡, and * indicates the $p$-value $< 0.0001$ in t-test.

| Model | diginetica | | | | | | Tmall | | | | | |
|---|---|---|---|---|---|---|---|---|---|---|---|---|
| | Hits@10 | NDCG@10 | MRR@10 | Hits@20 | NDCG@20 | MRR@20 | Hits@10 | NDCG@10 | MRR@10 | Hits@20 | NDCG@20 | MRR@20 |
| FPMC | 31.57* | 17.40* | 13.08* | 43.19* | 20.33* | 13.88* | 13.71* | 9.02* | 7.56* | 16.44* | 9.71* | 7.74* |
| GRU4Rec | 36.77* | 20.71* | 15.80* | 49.68* | 23.97* | 16.70* | 18.82* | 12.28* | 10.25** | 22.68* | 13.25* | 10.51* |
| NARM | 35.98* | 20.18* | 15.36* | 48.89* | 23.44* | 16.26* | 22.74* | 15.46* | 13.19* | 26.73* | 16.47* | 13.47* |
| STAMP | 33.59* | 18.89* | 14.41* | 45.87* | 22.00* | 15.26* | 24.32* | 16.55* | 14.12* | 28.40* | 17.58* | 14.41* |
| CSRM | 33.97* | 19.43* | 14.98* | 45.83* | 22.42* | 15.80* | 25.13* | 18.56* | 16.48* | 27.94* | 19.27* | 16.68* |
| S3-Rec‡ | 33.48* | 18.58* | 14.04* | 45.97* | 21.74* | 14.90* | 18.24* | 12.30* | 10.46* | 22.31* | 13.32* | 10.74* |
| M2TRec‡ | 29.67* | 16.30* | 12.23* | 41.23* | 19.22* | 13.02* | 11.42* | 7.56* | 6.36* | 13.75* | 8.15* | 6.52* |
| SR-GNN | 35.21* | 19.68* | 14.94* | 47.99* | 22.90* | 15.82* | 18.21* | 12.11* | 10.20* | 21.34* | 12.91* | 10.42* |
| GC-SAN | 35.25* | 19.72* | 14.97* | 47.87* | 22.90* | 15.85* | 19.29* | 12.80* | 10.78* | 23.18* | 13.78* | 11.05* |
| S2-DHCN | 30.76* | 17.04* | 12.86* | 42.39* | 19.98* | 13.66* | 22.00* | 13.36* | 10.68* | 27.23* | 14.69* | 11.05* |
| GCE-GNN | 36.32* | 20.77* | 16.02* | 48.67* | 23.89* | 16.87* | 28.33* | 20.01* | 17.32* | 30.24* | 20.50* | 17.45* |
| LESSR | 33.68* | 18.71* | 14.14* | 46.23* | 21.88* | 15.01* | 20.99* | 14.64* | 12.13* | 25.92* | 13.96* | 10.50* |
| MSGIFSR | 34.74* | 19.43* | 14.76* | 46.23* | 21.88* | 15.01* | 23.18* | 15.19* | 12.69* | 27.78* | 16.35* | 13.01* |
| FAPAT‡ | **37.42** | **21.31** | **16.39** | **50.41** | **24.59** | **17.29** | **32.45** | **22.02** | **18.72** | **36.18** | **22.97** | **18.99** |
| *Improv.* | *3.03%* | *2.60%* | *2.31%* | *1.46%* | *2.59%* | *2.49%* | *14.19%* | *10.04%* | *8.08%* | *19.64%* | *12.05%* | *8.83%* |

Table 3: Performance evaluation for next-item prediction on our 100 million industrial data.

| Model | Beauty | | | | Books | | | | Electronics | | | |
|---|---|---|---|---|---|---|---|---|---|---|---|---|
| | Hits@10 | NDCG@10 | Hits@20 | NDCG@20 | Hits@10 | NDCG@10 | Hits@20 | NDCG@20 | Hits@10 | NDCG@10 | Hits@20 | NDCG@20 |
| CSRM | 89.74 | 75.28 | 92.61 | 76.01 | 78.69 | 56.70 | 82.88 | 57.77 | 62.28 | 44.35 | 67.47 | 45.67 |
| S3-Rec‡ | 89.64 | 75.56 | 92.53 | 76.30 | 75.00 | 58.54 | 79.45 | 59.67 | 74.36 | 56.03 | 79.63 | 57.37 |
| M2TRec‡ | 80.13 | 65.97 | 83.66 | 66.87 | 32.56 | 22.58 | 35.39 | 25.70 | 57.32 | 44.84 | 61.70 | 45.95 |
| SR-GNN | 88.69 | 70.42 | 91.74 | 71.20 | 66.55 | 47.55 | 69.77 | 48.37 | 74.86 | 54.30 | 79.66 | 55.52 |
| GCE-GNN | 89.34 | 73.15 | 91.29 | 73.65 | 77.61 | 57.60 | 80.03 | 58.22 | 72.93 | 53.74 | 78.49 | 55.15 |
| LESSR | 89.95 | 71.29 | 92.98 | 72.06 | 73.72 | 53.86 | 82.31 | 54.77 | 72.91 | 50.46 | 78.78 | 51.96 |
| MSGIFSR | 90.18 | 73.62 | 92.50 | 74.21 | 72.93 | 52.23 | 76.33 | 53.09 | 73.56 | 53.83 | 77.45 | 54.73 |
| FAPAT‡ | **92.72** | **76.29** | **94.10** | **76.87** | **81.62** | **61.08** | **85.12** | **61.97** | **78.36** | **56.81** | **82.81** | **57.94** |
| *Improv.* | *2.82%* | *0.97%* | *1.20%* | *0.75%* | *3.72%* | *4.34%* | *2.70%* | *3.85%* | *4.68%* | *1.39%* | *3.95%* | *0.99%* |

assigns higher scores to hits at the top of the list. MRR@$K$ is the average of reciprocal ranks, with ranks above $K$ assigned 0. For public benchmarking, we report the average performance across seeds $\{2020, \cdots, 2024\}$, while for industrial data, we use a fixed seed (see Appendix E for more details).

## 5.2 Next-item Prediction Evaluation

**Experimental Results.** We first compare our models with selected baselines on two public datasets in Table 2. Overall, there is no huge difference between sequence and graph models when the session historical information is limited (e.g., in *diginetica*). However, we observe a performance boost with the history attention mechanism and session topology. FPMC that utilizes first-order Markov chains and matrix factorization is the worst. This reveals the difference between traditional recommendation and session-based recommendation. In contrast, RNN-based methods (GRU4Rec, NARM, STAMP, and CSRM) show better generalizability, along with the benefits of attention and memory. But we do not see further gains from pretraining and multi-task learning in S3-Rec and M2TRec. On the other hand, GNN-based algorithms for local sessions also achieve comparable results even with the absence of temporal signals. Moreover, the heterogeneous graphs of MSGIFSR beat the shortcut graphs from LESSR, indicating over-smoothing and potential noise. Although explicit global collaborative graphs enhance GCE-GNN, the implicit collaborative information from neurons and patterns in FAPAT easily surpasses other baselines significantly. At the same time, we evaluate algorithms on our 100 million industrial data. Table 3 demonstrates that most sequence models become unstable except S3-Rec. It employs incomplete data to predict masked items and attributes during pretraining and identifies contextualized collaborations by determining whether two incomplete sequences belong to the same session. But the multi-tasking learning for predicting attributes makes M2TRec hard to recommend items. While most GNN methods are competitive, S3-Rec and FAPAT are still better.

**Over-smoothing Relief.** We examine different graph topologies in baselines and FAPAT by comparing the graph density of different methods: the density of local session graphs (in SR-GNN), global collaborative filtering graphs(in GCE-GNN), shortcut graphs (in LESSR), heterogeneous graphs (in MSGIFSR), attribute patterns (in FAPAT) on E-commerce data are 3.658, 114.910, 25.447, 2.598, and 1.117, respectively. The density decrease significantly relieves the over-smoothing. Detailed analyses and expectations are provided in Appendix B.

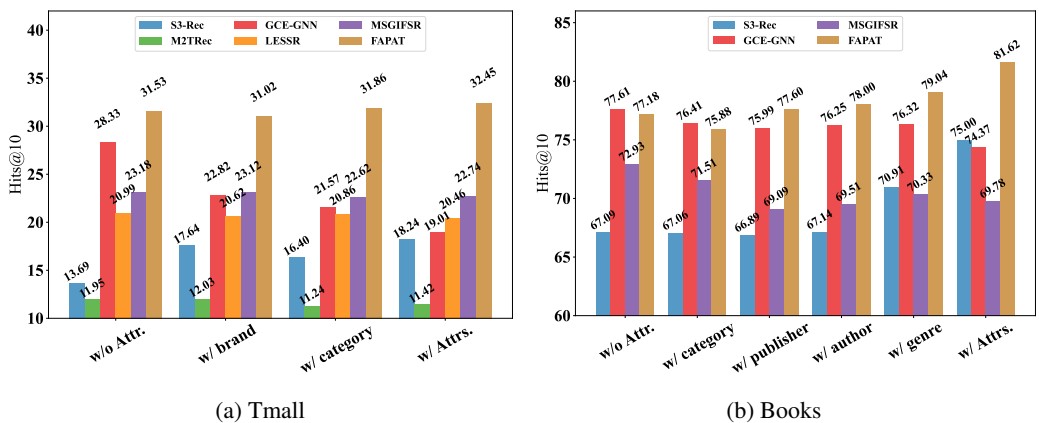

(a) Tmall         (b) Books

Figure 5: Effects of different attribute settings.

Table 4: Results of encoder comparison, where * indicates the p-value < 0.05 in t-test.

| Encoder | diginetica | | Tmall | | Beauty | | Books | |
|---|---|---|---|---|---|---|---|---|
| | Hits@10 | MRR@10 | Hits@10 | MRR@10 | Hits@10 | MRR@10 | Hits@10 | MRR@10 |
| FAPAT w/o Attr. | **36.82** | 16.29 | **31.53** | **18.86** | **88.70** | **67.11** | **77.18** | **49.66** |
| GraphFormer | 36.05* | 16.17 | 30.05* | 18.58 | 88.48 | 65.65 | 77.03 | 47.86 |
| Transformer | 36.30* | 16.02* | 28.83* | 18.30* | 88.10 | 65.90 | 74.67 | 46.54 |

## 5.3 Ablation Study

**Attribute Pattern Augmentation.** To evaluate the effect of attribute patterns, we conduct experiments on variants with single attributes or without any attribute. The same soft attention strategy from Eq. (6) is employed to fuse attribute embeddings for competitive baselines. Results in Figure 5 show that attribute pattern augmentation is more stable than attribute soft attention. FAPAT benefits from graph-nested attention, where the graph attention aligns the hidden space, and the memory attention captures item correlations and user intents. But attribute embeddings may have side effects on optimization in baselines, especially in graph neural networks. We also discover that not all attributes have a positive impact. Comparing among attribute pattern numbers, attribute patterns with significant frequencies (slightly lower than or similar to the item number) can have adverse effects.

**Graph-nested Attention.** The graph-nested attention is one of our contributions, distinguishing it from GraphFormer [41]. Unlike GraphFormer, our graph attention is integrated inside the blocks, which allows for direct benefit from the broader attention in the following self-attention via back-propagation. To ensure fairness, we replace the encoding module of FAPAT with GraphFormer and vanilla Transformer. Results in Table 4 demonstrate the advantages of our proposed graph-nested attention. Our experiments also show that even a simple vanilla Transformer can outperform previous state-of-the-art models by a significant margin, indicating the importance of emphasizing temporal information in SBRs and the appropriateness of attention for capturing long-distance dependencies.

## 5.4 Intent Capture Inspection

**Attribute Estimation.** Beyond item predictions, we also estimate the awareness of user intents from the product attribute side. We do not require models (except M2TRec) to predict attributes but to retrieve attributes from predicted items instead. We consider it a successful estimation if the top-ranked items have the same attribute value as the ground truth. M2TRec performs well on public data with multi-task learning but struggles on industrial E-commerce data with four attributes, as shown in Table 5. Pretraining in S3-Rec is not helpful due to catastrophic forgetting. GNN-based models perform similarly, except on *Tmall*, where the data are too sparse so that global collaborative information assists. On the contrary, FAPAT achieves robust predictions across four datasets. Even when session data are sufficient, frequent patterns remain effective.

Table 5: Attribute estimation evaluation.

| Model | *diginetica* | | *Tmall* | | *Beauty* | | *Books* | |
|---|---|---|---|---|---|---|---|---|
| | Hits@10 | MRR@10 | Hits@10 | MRR@10 | Hits@10 | MRR@10 | Hits@10 | MRR@10 |
| CSRM | 89.87* | 87.91* | 48.03* | 34.97* | 94.82 | 83.57 | 88.93 | 73.31 |
| M2TRec‡ | **94.10** | **89.76** | 59.44 | 39.72 | 95.93 | 84.95 | 83.80 | 71.22 |
| S3-Rec‡ | 89.66* | 88.42* | 44.95* | 32.60* | 95.35 | 84.79 | 88.11 | 75.63 |
| GCE-GNN | 89.89* | 87.96* | 55.88* | 38.57* | 95.44 | 84.16 | 89.78 | 75.68 |
| LESSR | 88.60* | 86.24* | 49.43* | 32.61* | 95.02 | 81.56 | 89.57 | 73.73 |
| MSGIFSR | 89.48* | 87.19* | 50.09* | 33.95* | 95.79 | 85.42 | 86.67 | 71.99 |
| FAPAT‡ | 89.99* | 88.23* | **59.49** | **40.56** | **95.94** | **86.87** | **90.77** | **76.83** |
| w/o Attr. | 89.35* | 87.50* | 58.22* | 40.08 | 95.07 | 81.65 | 89.29 | 74.12 |

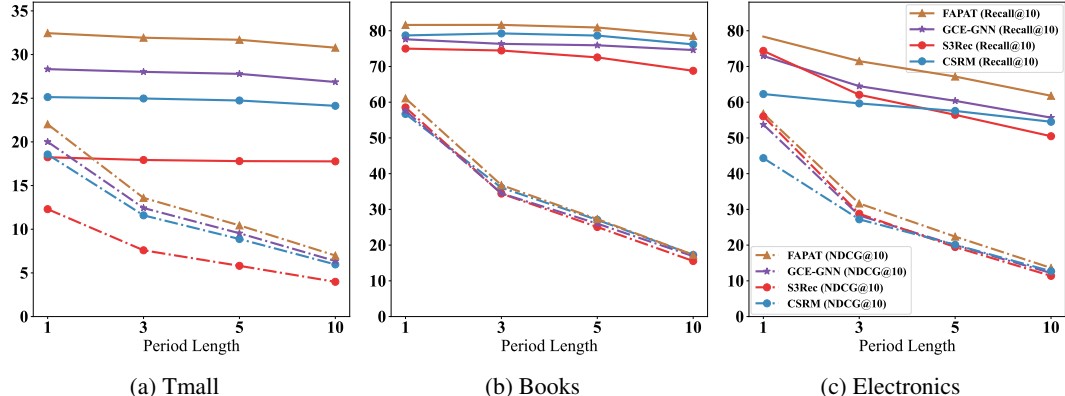

(a) Tmall          (b) Books          (c) Electronics

Figure 6: Period-item recommendation evaluation.

**Period-item Recommendation.** In addition to single-step evaluation, we revisit recommenders. A sound and robust SRB system must understand user intents in deep and foresee the possible consistency and the potential change. Autoregressive settings pose challenges for GNNs due to short click histories and error accumulation. Therefore, we evaluate period recommendations like search engines by comparing the top-10 predicted items with the next 3/5/10 clicks. Figure 6 demonstrates period-recommendation performance. Sequence models offer steady results in Recall, indicating that temporal information is one of the prerequisites to analyzing users' latest intents. However, pretraining may hinder models' ability to generalize over long periods, resulting in a severe decline in NDCG. Our FAPAT achieves the best performance in all metrics, indicating the effectiveness of attribute graphlets in capturing deep user intents.

## 6 Conclusion

Our paper introduces FAPAT, a novel framework that leverages attribute graph patterns to augment anonymous sequence encoding for session-based recommendations. Compared to other GNN-based methods, frequent attribute graphlets can reduce noise and topology densities for enhancing user intent capture. Our sequence encoder can better preserve temporal signals and forecast the user's latest intents. Experimental results clearly illustrate the effectiveness, and extensive ablation studies and intent capture inspections provide additional support. We discuss limitations in Appendix A. One of the future works is to improve ranking priority by combining pretraining and pattern augmentation.

## Acknowledgments

The authors of this paper were supported by the NSFC Fund (U20B2053) from the NSFC of China, the RIF (R6020-19 and R6021-20) and the GRF (16211520 and 16205322) from RGC of Hong Kong. We also thank the support from the UGC Research Matching Grants (RMGS20EG01-D, RMGS20CR11, RMGS20CR12, RMGS20EG19, RMGS20EG21, RMGS23CR05, RMGS23EG08).

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

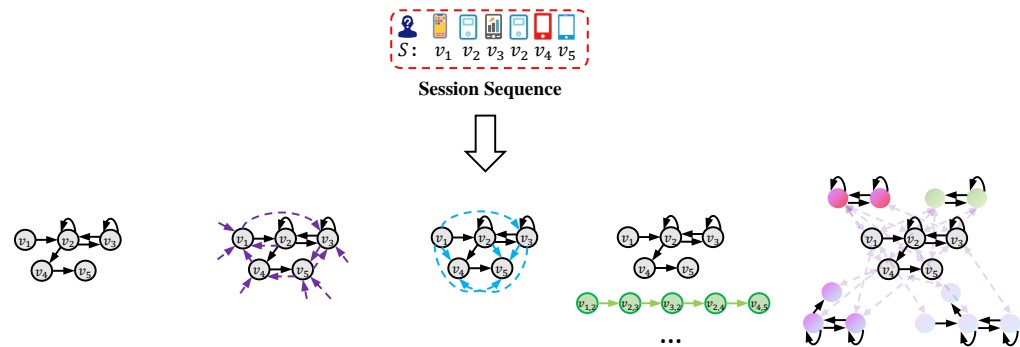


Local Session Graph   Global Transition Graph   Shortcut Graph   Heterogeneous Graphs   Attribute Patterns


Figure 7: Different granularities of graphs derived from a session, where the violet edges, blue edges, and green edges correspond to the global collaborations, shortcuts, and higher-level heterogeneous connections, respectively.

## A  Limitations

There are three limitations to our current proposed method and evaluation. First, our method separately processes and retrieves patterns for each attribute type. We do not merge all attributes in a candidate pool because we aim for our method to easily generalize to real recommendation systems with hundreds of attribute types and category hierarchies. The current implementation supports adding a new attribute type to a model as long as its embeddings align with the embeddings of other attributes. Second, we conducted experiments based on "clean" session data. Most E-commerce platforms do not have truly clean data on product attributes, so attribute data, in general, is very sparse and full of invalid values. We performed human-centric attribute regularization to drop products without valid attribute values, which may create a gap compared to a real industrial system. Third, the evaluation does not consider the same products with different identifiers. Therefore, evaluating results (especially MRR) cannot accurately reflect the performance. To better reflect the real performance with error tolerance, a larger K is suggested. The current comparison is still fair for all algorithms, and we address this synonym problem in attribute estimation in § 5.4, where we merge attribute values based on semantics and syntax.

## B  Transition Graph Density

The graph structure is crucial for neural networks to capture explicit transitions and implicit connections. A local session records the history of a user's clicks or purchases, which is usually sparse. In contrast, the global collaborative graph could be extremely dense because each pair of items may have a potential connection. From this perspective, the density indicates the explicit information provided from session data. On the other hand, different graph topologies and densities also present different focuses and challenges. The sparse local transition graph emphasizes current intents, while the global collaboration indicates broader interests and revenues. Graph neural networks excel at capturing local features, but a large number of neighbors can overshadow important connections with less significant ones. Considering that previous methods have focused on different granularities individually, we summarize them in Figure 7 and compare them in terms of optimization interpretations[3].

- **Local** session graphs correspond to item transitions within a session, where edges are created between two consecutively clicked/purchased items. The density is usually sparse (slightly greater than 1.0), allowing exploration and global collaboration to be learned through model parameters instead of explicit connections. Therefore, generalizing to unseen click patterns becomes challenging.

- **Global** transition graphs record all collaborations. In a real industrial system, the density is usually beyond one hundred or even one thousand. Ideally, any session can benefit from this global collaborative information, including multi-hop connections. However, optimizing graph neural

---

[3]Graphs are typically considered undirected in practical algorithms.

networks to learn such topologies (due to oversmoothing) and building a large model for real-time inference (due to latency and streaming processing) pose challenges.

- **Shortcut** graphs aim to avoid constructing global graphs and make session learning more efficient. They were proposed by LESSR [1] to address information loss in graph convolutions. Specifically, they allow latent items to be aware of all previous clicks, resembling shortcuts for multi-hop neighbors in the directed local session graph. However, they lack the extensive exploration capabilities of the global transition graph and suffer from oversmoothing issues due to dense connections.

- **Heterogeneous** graphs strike a balance between shortcuts and local adjacency. Nodes with different numbers of items are categorized into different groups, and transition edges capture varying levels of spatial continuity. From a high-level perspective, this graph is sparser than the local session graph, resulting in faster convergence for optimization. However, the propagation of high-order information introduces additional processing costs and the risk of overfitting.

- **Patterns**, especially attribute patterns, should be the most efficient features for recommendations in a large candidate item pool. Each pattern can be considered a higher-grained heterogeneous graph. However, pattern filtering can significantly eliminate noise influence, not to mention the benefits gained from offline indexing. Besides, the partial match of patterns can provide the intent information from other sessions, making the learning and prediction more reliable and steady.

## C   Experimental Data

### C.1   Public Benchmarks

We choose two public benchmarks for session-based recommendation evaluation: *diginetica* [4] is CIKM Cup 2016 that contains the browser logs and anonymized transactions; *Tmall* [5] comes from a competition in IJCAI-15 which collects anonymous users' shopping logs on the Tmall online website. We acquire attributes from the original data and drop items without attributes or with invalid values. Therefore, the performance of baselines may not be exactly same as the reported numbers in the original papers.

### C.2   E-commerce Data Collection

We collect E-commerce data from our log systems in two months. We follow the same procedure to clean and process session data in *beauty*, *books*, and *electronics* domains[6]:

  I   We focus on successful purchases so that we only keep sessions ending with "purchase" actions.

 II   To make sure previous clicks can reflect the purchase intent, we drop actions 20 minutes ago.

III   We filter out items with missing attributes (i.e., books without publishers, authors, or genre, and electronics without colors and brands).

IV   We adopt the 20-core setting to finalize the item sets, in which items appear on at least 20 different days.

 V   Only sessions whose length is no greater than 50 are preserved.

VI   We retrieve item attributes in our attribute databases.

VII   For GNN models that requires the global transition graph from training data, we maintain 12 neighbors based on the co-occurrence, which is consistent with GCE-GNN [34].

---

[4]https://competitions.codalab.org/competitions/11161
[5]https://tianchi.aliyun.com/dataset/dataDetail?dataId=42
[6]The sampled data scales and distributions are different in real systems due to out-of-domain items filtering.

Table 6: Statistics of datasets based on timestamps.

| | Public | | Industrial (E-commerce) | | |
| | diginetica | Tmall | Beauty | Books | Electronics |
|---|---|---|---|---|---|
| #User | 57,623 | 7,576 | 2.6 M | 3.2 M | 10.2 M |
| #Item | 43,074 | 39,768 | 39.2 K | 94.8 K | 244.7 K |
| #Click | 993,163 | 438,315 | 27.2 M | 38.8 M | 115.6 M |
| Avg. Len. | 4.850 | 6.649 | 10.325 | 11.912 | 11.249 |
| #Train | 630,789 | 303,181 | 19.6 M | 28.2 M | 84.1 M |
| #Valid | 78,708 | 33,735 | 2.4 M | 3.5 M | 10.5 M |
| #Test | 78,907 | 35,481 | 2.5 M | 3.8 M | 10.6 M |
| #Attribute | category : 995 | category : 821
brand : 4,304 | category : 359
color : 1,101
brand : 4,359
size : 1,883 | category : 18
publisher : 2,751
author : 27,651
genre : 2,634 | type : 123
category : 881
color : 2,096
brand : 24,196 |
| #Pattern | category : 1,866 | category : 33,582
brand : 2,497 | category : 970
color : 4,059
brand : 254
size : 1,091 | category : 24
publisher : 4,370
author : 1,399
genre : 12,535 | type : 9,289
category : 13,991
color : 146,402
brand : 14,043 |
| Density | Local: 0.886
Global: 11.329
Shortcut: 2.512
Heterogeneous: 0.543
Pattern: 1.023 | Local: 1.249
Global: 10.222
Shortcut: 4.983
Heterogeneous: 0.707
Pattern: 1.165 | Local: 4.510
Global: 70.504
Shortcut: 29.827
Heterogeneous: 3.412
Pattern: 1.095 | Local: 3.554
Global: 99.389
Shortcut: 26.649
Heterogeneous: 2.333
Pattern: 1.085 | Local: 2.910
Global: 128.041
Shortcut: 19.865
Heterogeneous: 2.049
Pattern: 1.189 |

## C.3 Data Split

We follow previous settings that split training/validation/testing data based on timestamps. For *diginetica*, we gather the last 8-14 days as validation, the last 7 days as testing, and remaining as training. For *Tmall*, we use the last 101-200 seconds as validation, the last 100 seconds as testing, and remaining as training. For our industrial E-commerce data (i.e., *Beauty*, *Books*, *Electronics*), we select the last 6-10 days as validation, the last 5 days as testing, and remaining as training.

## C.4 Data Statistics

Table 6 summarizes the statistics of the experimental datasets based on timestamps. The density is calculated based on undirected graphs, which would be doubled during graph convolution in practice. *Local density*, as used in SR-GNN and GC-SAN, corresponds to the average density of local session graphs in E-commerce sessions. On the other hand, *global density*, as used in GCE-GNN, refers to the density of the global collaborative graph obtained by connecting all adjacent items appearing in all sessions. *Shortcut density*, as used in LESSR, is the density resulting from connecting all items in a single session as a complete graph. *Heterogeneous density*, as used in MSGIFSR, refers to the average density of the heterogeneous graphs obtained by regarding the consecutive adjacent two nodes as a fine-grained intent unit. Lastly, *pattern density*, as used in FAPAT, is the density of the acquired frequent and compact patterns. From Table 6, it is evident that leveraging patterns is the most effective way of characterizing user intents because other graph topologies vary with data sources and scales, making it difficult to generalize and provide stable performance. Besides, patterns can be preprocessed as indicies to aid recommendations, making them more practical in industrial scenarios. Moreover, it is easy to update attribute patterns dynamically, whereas other graph structures are more closely coupled with input sessions and are more sensitive to tiny variations.

## D   Baselines

We compare our method with the following baselines:
*Sequence-based methods*

- **FPMC** [27] learns the representation of session via Markov-chain based methods.
- **GRU4Rec** [8] is the first RNN-based approach that simulates the Markov Decision Process (MDP) but has a better generalization.
- **NARM** [16] is a attention-based RNN model to learn session embeddings.
- **STAMP** [19] adopts attention mechanism between the last item to previous histories to represent users' short-term interests.

- **CSRM** [31] proposes to engage an inner memory encoder and external memory network to capture correlations between neighborhood sessions to enrich the collaborative representations.
- **S3-Rec** [47] is the first pretrained SBR model that predicts items, attributes, and segments during the pretraining stage.
- **M2TRec** [28] is a metadata-aware multi-task Transformer model. In the original paper, the authors ignore item embeddings. For a fair comparison, we also regard the item ids as one of metadata.

*Graph-based methods*

- **SR-GNN** [35] is the first GNN-based model for the SBR task, which transforms the session data into a direct unweighted graph and learns the representation of the item-transitions graph.
- **GC-SAN** [38] uses gated GNNs to extract local context information and then self-attention to obtain the global representation.
- **S2-DHCN** [37] transforms the session data into hyper-graphs and line-graphs and encodes them via GCNs to enhance the session representations.
- **GCE-GNN** [34] aggregates two levels of item embeddings from session graphs and global graphs with soft attention.
- **LESSR** [1] preserves the edge order and constructs shortcuts to encode sessions for GNNs.
- **MSGIFSR** [7] captures the user intents from multiple granularities to relieve the computational burden of long-dependency. In experiments, we search the best model from the level-1, level-2, and level-3 consecutive intent units.

## E  Experimental Settings

We fix all embeddings and hidden dimensions as 100, and the batch size is searched among {100, 200, 500} for all methods. We also choose the number of layers/iterations (if applicable) from the validation performance (e.g., MRR@10). A learning scheduler with 10% linear warmup and 90% decay is associated with the Adam optimizer [14]. The initial learning rate is set as 1e-3, and the regularization weight is tuned among {1e-4, 1e-5, 1e-6}. We seek the dropout probability between two modules from {0.0, 0.2, 0.4}, but fix the attention dropout rate as 0.2. The number of attention heads is empirically set as 4. We follow the setting of GCE-GNN that the maximum one-hop neighbor number in GAT is 12. In the interest of fairness, we also set the maximum selected pattern number as 12. Hyper-parameter tuning is time costly on our industrial data so that we use the best combinations obtained from one day transactions. We implement our methods and run experiments with Python and PyTorch over 8 x A100 NVIDIA GPUs.

## F  Experimental Results

Due to the space limit, we only report some results in the main content. More comprehensive comparisons are shown in Tables 7-11, where standard deviations are enclosed in brackets. The best and second-best results are respectively highlighted in bold and underlined. Methods that use attributes are marked with ‡, and * indicates the $p$-value $< 0.0001$ in t-test.

Table 7: Performance evaluation for next-item prediction on *diginetica*.

| Model | diginetica | | | | | |
|---|---|---|---|---|---|---|
| | Hits@10 | NDCG@10 | MRR@10 | Hits@20 | NDCG@20 | MRR@20 |
| FPMC | 31.57(0.04)* | 17.40(0.01)* | 13.08(0.02)* | 43.19(0.05)* | 20.33(0.03)* | 13.88(0.03)* |
| GRU4Rec | 36.77(0.14)* | 20.71(0.05)* | 15.80(0.03)* | 49.68(0.06)* | 23.97(0.03)* | 16.70(0.03)* |
| NARM | 35.98(0.10)* | 20.18(0.06)* | 15.36(0.06)* | 48.89(0.12)* | 23.44(0.06)* | 16.26(0.06)* |
| STAMP | 33.59(0.15)* | 18.89(0.18)* | 14.41(0.19)* | 45.87(0.15)* | 22.00(0.18)* | 15.26(0.19)* |
| CSRM | 33.97(0.08)* | 19.43(0.03)* | 14.98(0.03)* | 45.83(0.02)* | 22.42(0.02)* | 15.80(0.02)* |
| S3-Rec‡ | 33.48(0.13)* | 18.58(0.09)* | 14.04(0.10)* | 45.97(0.08)* | 21.74(0.09)* | 14.90(0.10)* |
| M2TRec‡ | 29.67(0.43)* | 16.30(0.24)* | 12.23(0.18)* | 41.23(0.63)* | 19.22(0.29)* | 13.02(0.20)* |
| SR-GNN | 35.21(0.02)* | 19.68(0.04)* | 14.94(0.04)* | 47.99(0.04)* | 22.90(0.04)* | 15.82(0.04)* |
| GC-SAN | 35.25(0.09)* | 19.72(0.04)* | 14.97(0.03)* | 47.87(0.09)* | 22.90(0.04)* | 15.85(0.03)* |
| S2-DHCN | 30.76(0.07)* | 17.04(0.14)* | 12.86(0.16)* | 42.39(0.07)* | 19.98(0.13)* | 13.66(0.16)* |
| GCE-GNN | 36.32(0.09)* | 20.77(0.07)* | 16.02(0.07)* | 48.67(1.12)* | 23.89(0.23)* | 16.87(0.03)* |
| LESSR | 33.68(0.05)* | 18.71(0.03)* | 14.14(0.03)* | 46.23(0.11)* | 21.88(0.05)* | 15.01(0.03)* |
| MSGIFSR | 34.74(0.09)* | 19.43(0.06)* | 14.76(0.07)* | 46.23(0.11)* | 21.88(0.05)* | 15.01(0.03)* |
| FAPAT‡ | **37.42**(0.10) | **21.31**(0.03) | **16.39**(0.04) | **50.41**(0.15) | **24.59**(0.06) | **17.29**(0.04) |
| *Improv.* | *3.03%* | *2.60%* | *2.31%* | *1.46%* | *2.59%* | *2.49%* |

Table 8: Performance evaluation for next-item prediction on *Tmall*.

| Model | Tmall | | | | | |
|---|---|---|---|---|---|---|
| | Hits@10 | NDCG@10 | MRR@10 | Hits@20 | NDCG@20 | MRR@20 |
| FPMC | 13.71(0.16)* | 9.02(0.02)* | 7.56(0.03)* | 16.44(0.23)* | 9.71(0.04)* | 7.74(0.02) |
| GRU4Rec | 18.82(0.17)* | 12.28(0.11)* | 10.25(0.09)* | 22.68(0.21)* | 13.25(0.12)* | 10.51(0.10)* |
| NARM | 22.74(0.20)* | 15.46(0.12)* | 13.19(0.10)* | 26.73(0.26)* | 16.47(0.13)* | 13.47(0.10)* |
| STAMP | 24.32(0.31)* | 16.55(0.29)* | 14.12(0.29)* | 28.40(0.35)* | 17.58(0.30)* | 14.41(0.29)* |
| CSRM | 25.13(0.19)* | 18.56(0.18)* | 16.48(0.18)* | 27.94(0.15)* | 19.27(0.17)* | 16.68(0.18)* |
| S3-Rec‡ | 18.24(0.11)* | 12.30(0.07)* | 10.46(0.06)* | 22.31(0.17)* | 13.32(0.08)* | 10.74(0.06)* |
| M2TRec‡ | 11.42(0.21)* | 7.56(0.06)* | 6.36(0.11)* | 13.75(0.35)* | 8.15(0.04)* | 6.52(0.10)* |
| SR-GNN | 18.21(0.51)* | 12.11(0.32)* | 10.20(0.28)* | 21.34(0.49)* | 12.91(0.31)* | 10.42(0.28)* |
| GC-SAN | 19.29(0.14)* | 12.80(0.07)* | 10.78(0.13)* | 23.18(0.23)* | 13.78(0.04)* | 11.05(0.12)* |
| S2-DHCN | 22.00(0.36)* | 13.36(0.21)* | 10.68(0.17)* | 27.23(0.33)* | 14.69(0.20)* | 11.05(0.17)* |
| GCE-GNN | 28.33(0.13)* | 20.01(0.12)* | 17.32(0.13)* | 30.24(0.16)* | 20.50(0.13)* | 17.45(0.13)* |
| LESSR | 20.99(0.26)* | 14.64(0.18)* | 12.13(0.19)* | 25.92(0.23)* | 13.96(0.22)* | 10.50(0.23)* |
| MSGIFSR | 23.18(0.19)* | 15.19(0.11)* | 12.69(0.10)* | 27.78(0.25)* | 16.35(0.11)* | 13.01(0.09)* |
| FAPAT‡ | **32.45**(0.21) | **22.02**(0.15) | **18.72**(0.13) | **36.18**(0.21) | **22.97**(0.14) | **18.99**(0.13) |
| *Improv.* | *14.19%* | *10.04%* | *8.08%* | *19.64%* | *12.05%* | *8.83%* |

Table 9: Performance evaluation for next-item prediction on *Beauty*.

| Model | Beauty | | | | | |
|---|---|---|---|---|---|---|
| | Hits@10 | NDCG@10 | MRR@10 | Hits@20 | NDCG@20 | MRR@20 |
| FPMC | 72.00 | 57.20 | 52.42 | 75.91 | 58.19 | 52.70 |
| GRU4Rec | 73.95 | 58.19 | 53.13 | 78.54 | 59.36 | 53.45 |
| NARM | 88.09 | 70.44 | 64.68 | 91.50 | 71.31 | 64.93 |
| STAMP | 80.08 | 63.76 | 58.47 | 83.84 | 64.72 | 58.73 |
| CSRM | 89.74 | 75.28 | 70.56 | 92.61 | 76.01 | 70.77 |
| S3-Rec‡ | 89.64 | 75.56 | 70.99 | 92.53 | 76.30 | 71.19 |
| M2TRec‡ | 80.13 | 65.97 | 61.65 | 83.66 | 66.87 | 61.65 |
| SR-GNN | 88.69 | 70.42 | 64.44 | 91.74 | 71.20 | 64.65 |
| GC-SAN | 86.67 | 70.80 | 64.71 | 88.98 | 72.50 | 65.97 |
| S2-DHCN | 7.25 | 5.38 | 4.80 | 8.87 | 5.79 | 4.91 |
| GCE-GNN | 89.34 | 73.15 | 67.80 | 91.29 | 73.65 | 67.94 |
| LESSR | 89.95 | 71.29 | 65.18 | 92.98 | 72.06 | 65.40 |
| MSGIFSR | 90.18 | 73.62 | 65.18 | 92.50 | 74.21 | 65.65 |
| FAPAT‡ | **92.72** | **76.29** | **71.09** | **94.10** | **76.87** | **71.24** |
| *Improv.* | *2.82%* | *0.97%* | *0.14%* | *1.20%* | *0.75%* | *0.07%* |

Table 10: Performance evaluation for next-item prediction on *Books*.

| Model | Books | | | | | |
|---|---|---|---|---|---|---|
| | Hits@10 | NDCG@10 | MRR@10 | Hits@20 | NDCG@20 | MRR@20 |
| FPMC | 36.51 | 24.32 | 20.49 | 41.90 | 25.69 | 20.87 |
| GRU4Rec | 47.21 | 31.86 | 27.02 | 53.55 | 33.47 | 27.46 |
| NARM | 76.09 | 54.22 | 47.13 | 80.83 | 55.43 | 47.36 |
| STAMP | 61.49 | 42.13 | 35.95 | 67.46 | 43.65 | 36.37 |
| CSRM | 78.69 | 56.70 | 49.54 | 82.88 | 57.77 | 49.83 |
| S3-Rec‡ | 75.00 | 58.54 | 53.23 | 79.45 | 59.67 | 53.55 |
| M2TRec‡ | 32.56 | 22.58 | 24.98 | 35.39 | 25.70 | 22.78 |
| SR-GNN | 66.55 | 47.55 | 41.32 | 69.77 | 48.37 | 41.55 |
| GC-SAN | 72.56 | 54.92 | 49.25 | 75.73 | 56.05 | 50.14 |
| S2-DHCN | 4.69 | 3.42 | 3.03 | 5.60 | 3.65 | 3.09 |
| GCE-GNN | 77.61 | 57.60 | 51.00 | 80.03 | 58.22 | 51.17 |
| LESSR | 73.72 | 53.86 | 47.36 | 82.31 | 54.77 | 47.61 |
| MSGIFSR | 72.93 | 52.23 | 45.66 | 76.33 | 53.09 | 45.66 |
| FAPAT‡ | **81.62** | **61.08** | **54.39** | **85.12** | **61.97** | **54.64** |
| *Improv.* | *3.72%* | *4.34%* | *2.18%* | *2.70%* | *3.85%* | *2.04%* |

Table 11: Performance evaluation for next-item prediction on *Electronics*.

| Model | Electronics | | | | | |
|---|---|---|---|---|---|---|
| | Hits@10 | NDCG@10 | MRR@10 | Hits@20 | NDCG@20 | MRR@20 |
| FPMC | 37.87 | 26.91 | 23.42 | 42.07 | 27.97 | 23.71 |
| GRU4Rec | 58.46 | 40.69 | 35.02 | 64.42 | 42.21 | 35.44 |
| NARM | 61.10 | 41.20 | 32.05 | 77.36 | 44.56 | 33.75 |
| STAMP | 59.30 | 42.04 | 36.53 | 67.94 | 45.07 | 36.97 |
| CSRM | 62.28 | 44.35 | 38.59 | 67.47 | 45.67 | 38.96 |
| S3-Rec‡ | 74.36 | 56.03 | **50.16** | 79.63 | 57.37 | **50.53** |
| M2TRec‡ | 57.32 | 44.84 | 40.85 | 61.70 | 45.95 | 41.15 |
| SR-GNN | 74.86 | 54.30 | 47.66 | 79.66 | 55.52 | 48.00 |
| GC-SAN | 72.76 | 53.37 | 45.98 | 77.34 | 46.34 | 49.91 |
| S2-DHCN | 4.18 | 2.65 | 2.18 | 5.08 | 2.88 | 2.24 |
| GCE-GNN | 72.93 | 53.74 | 47.59 | 78.49 | 55.15 | 47.98 |
| LESSR | 72.91 | 50.46 | 43.26 | 78.78 | 51.96 | 43.67 |
| MSGIFSR | 73.56 | 53.83 | 47.77 | 77.45 | 54.73 | 48.02 |
| FAPAT‡ | **78.36** | **56.81** | 49.80 | **82.81** | **57.94** | 50.12 |
| *Improv.* | *4.68%* | *1.39%* | *-0.07%* | *3.95%* | *0.99%* | *-0.81%* |

