# OpenReview forum: "Enhancing User Intent Capture in Session-Based Recommendation with Attribute Patterns"
_NeurIPS.cc/2023/Conference — NeurIPS 2023 poster_

### Official Review · Reviewer_LurS · 2023-07-03

**Soundness:** 4 excellent
**Presentation:** 3 good
**Contribution:** 3 good
**Rating:** 7
**Confidence:** 4

**Summary:**

The paper presents a transformer-based method, namely Frequent Attribute Pattern Augmented Transformer (FAPAT), that considers attribute patterns as supplementary information. More specifically, FAPAT builds attribute transition graphs, mine frequent attribute patterns, and match attribute patterns to better captures user intents. Experiments conducted on two public benchmarks as well as three industrial datasets show that the proposed method deliver noticeable performance boosts (4.5%) averaged over all experiments. The code and datasets are promised to be released after acceptance.

**Strengths:**

1. Introduction section is written very well to present the problem and the high-level picture of the proposed method.
2. Table 1 in Related Work section is very straight-forward and efficient in comparing a large number of existing methods and demonstrating how they differ from the proposed method. And the categorical items (temporal info, history attention, etc.) are reasonably defined.
3. Methodology section is clear and easy to follow.
4. Experiments are thorough with a good number of baselines.

**Weaknesses:**

1. Some of the thought process is not explained, for example, in section 4.1.1, what are the thoughts process behind adopting gSpan and how were those twenty types of patterns chosen?

**Questions:**

1. In introduction, can you elaborate more on “We argue that the current use of item-side metadata provides little assistance in SBR models as user intent may change over time.“?
2. Typo in line 131 - "whi".
3. I am not sure what is the purpose of Section 3 - Background and Motivations, especially 3.2 and 3.3, they read like they should belong to Related Work.

**Limitations:**

Yes, the authors discuss limitations in Appendix A.

---

> ### Author Rebuttal · Authors · 2023-08-10
>
> **W1. Clarifications for pattern acquisition**
>
> A1. End-to-end retrieval becomes a viable option in the absence of resource constraints. However, given the multitude of substructures—numbering in the millions or even billions—within industrial sessions, we must ponder efficiency. Data mining methods, like gSpan, offer near-linear complexity, drastically trimming the pool of pattern candidates to a few thousand. This ensures efficiency and effectiveness in both training and inference. We exclusively consider motifs with three or four nodes, encompassing only those exhibiting circular or triangular structures. These two types of structures notably curtail randomness and enhance robustness.

---

### Official Review · Reviewer_WQyi · 2023-07-05

**Soundness:** 2 fair
**Presentation:** 2 fair
**Contribution:** 1 poor
**Rating:** 3
**Confidence:** 4

**Summary:**

This paper proposes FAPAT, which augments session-based recommendation models using item attributes. Specifically, FAPAT constructs attribute graphs and models user intent using pattern mining and an improved graph attention mechanism.

**Strengths:**

1. The paper studies an important application task, i.e., session-based recommendation.
2. Experiments are conducted on 2 public datasets and 3 large-scale industrial datasets.

**Weaknesses:**

1. Over-complicated method. Session-based recommendation methods that incorporate session graphs have been criticized for being overly complex. The significant increase in algorithm complexity generally yields limited improvements in performance. The method proposed in this paper, involving modules such as motif extraction and graph attention mechanisms, adds further complexity to this paradigm. However, based on the experiments in Table 2, even with such complexity, this method only outperforms the concise baseline GRU4Rec by a limited margin on the classic benchmark dataset diginetica. The results raise doubts about the necessity of such a complex approach. It could be possible to achieve comparable results by introducing attributes using a much simpler method.
2. Missing baselines. Two important baselines are missed. SASRec [1] is a highly popular baseline method for sequential/session-based recommendation, based on Transformer modules. FDSA [2] is a straightforward method based on SASRec that introduces item attributes. Both of these methods are concise and can be regarded as a basic version of the proposed model. These two methods should be included in experiments and compared against the proposed method.
3. Code is not available during the reviewing phase, making reproduction of the reported results difficult.

[1] Kang et al. Self-Attentive Sequential Recommendation. ICDM 2018.

[2] Zhang et al. Feature-level Deeper Self-Attention Network for Sequential Recommendation. IJCAI 2019.

**Questions:**

Please refer to "Weakness".

---

> ### Author Rebuttal · Authors · 2023-08-10
>
> **W1. Over-complicated method**
>
> A1. Differing Perspective on GRU4Rec. I must respectfully disagree with the reviewer's perspective. The counter-example of GRU4Rec showcases remarkable performance on diginetica (mean length: 4.850). Nevertheless, its effectiveness significantly wanes on Tmall (mean length: 6.649) and our extensive industrial dataset (mean length exceeding 10). This results in substantial disparities (18.82 vs. 32.45, 73.95 vs. 92.72, 47.21 vs. 81.62, and 58.46 vs. 78.36). Such variations provide robust validation for the ingenuity and novelty driving our proposed approach.
>
> **W2. Missing baselines**
>
> A2: We extend our gratitude to the reviewer for highlighting the two baselines. We will comprehensively discuss these baselines in the final version. Regarding SASRec [1], which employs stacked Transformer layers, it can be seen as the non-pretrained S3-Rec model. The performance comparison is presented below:
> | method | | diginetica | | | Tmall | | | Beauty | | | Books | | | Electronics | |
> | --- | --- | --- | --- | --- | --- | --- | --- | --- | --- | --- | --- | --- | --- | --- | --- |
> | | Recall@10 | NDCG@10 | MRR@10 | Recall@10 | NDCG@10 | MRR@10 | Recall@10 | NDCG@10 | MRR@10 | Recall@10 | NDCG@10 | MRR@10 | Recall@10 | NDCG@10 | MRR@10 |
> SASRec | 32.15 | 17.86 | 13.52 | 13.69 | 9.47 | 8.16 | 85.18 | 70.87 | 66.21 | 67.09 | 52.10 | 47.25 | 61.14 | 45.31 | 40.08 |
> S3Rec | 33.48 | 18.58 | 14.04 | 18.24 | 12.30 | 10.46 | 89.64 | 75.56 | 70.99 | 75.00 | 58.54 | 53.23 | 74.36 | 56.03 | 50.16 |
> FAPAT | 37.42 | 21.31 | 16.39 | 32.45 | 22.02 | 18.72 | 92.72 | 76.29 | 71.09 | 81.62 | 61.08 | 54.39 | 78.36 | 56.81 | 49.80 | FDSA [2] introduces separate channels for features, demonstrating the effectiveness of metadata. We are unable to present the results due to data security concerns. However, based on our experience, FDSA's performance lies between SASRec and S3-Rec. A comprehensive discussion of these two baselines will be included in the final version.
>
> **W3. Unavailable code**
>
> A3. Code can't be uploaded due to security concerns currently. It'll be released upon paper acceptance and security department's approval.

---

> > ### Comment · Reviewer_WQyi · 2023-08-18
> > **Thank you for the constructive rebuttal**
> >
> > Thank you for the additional details provided in response to my comments.
> >
> > **Reply to W2. Missing baselines - (1)**
> >
> > However, I believe it is important to address all concerns raised. Unfortunately, there's still a significant issue that hasn't been adequately addressed: the absence of a comparison with the FDSA baseline in your experiments. FDSA [2], which introduces item attributes into the SASRec, sharing the same task formulation and can be seen as a basic form of your model. It is therefore imperative to include it for a comprehensive performance evaluation.
> >
> > As previously mentioned, I'm concerned about the increased complexity of your proposed method. FDSA might help to test whether comparable performance can be achieved using a simpler method to introduce item attributes. Thus, I encourage you to include a comparison against FDSA in your final version.
> >
> > **Reply to W2. Missing baselines - (2)**
> >
> > In addition to the aforementioned comments regarding the missing baseline method FDSA, I also find the supplementary results you provided for SASRec to be unusual.
> >
> > It is generally accepted within the community that SASRec usually outperforms GRU4Rec on most datasets, as demonstrated in papers such as BERT4Rec [3], S^3-Rec [4], TiSASRec [5], CL4Rec [6], and HyperRec [7]. However, in the tables you've provided as a supplement, SASRec appears to underperform in comparison to GRU4Rec. This is notably uncommon and raises concerns regarding the reliability and validity of the results presented.
> >
> > I would greatly appreciate it if you could clarify these discrepancies, as proper baseline comparison is crucial for contextualizing the effectiveness of the proposed method.
> >
> > [2] Zhang et al. Feature-level Deeper Self-Attention Network for Sequential Recommendation. IJCAI 2019.
> >
> > [3] Sun et al. BERT4Rec: Sequential Recommendation with Bidirectional Encoder Representations from Transformer. CIKM 2019.
> >
> > [4] Zhou et al. S^3-Rec: Self-Supervised Learning for Sequential Recommendation with Mutual Information Maximization. CIKM 2020.
> >
> > [5] Li et al. Time Interval Aware Self-Attention for Sequential Recommendation. WSDM 2020.
> >
> > [6] Xie et al. Contrastive Learning for Sequential Recommendation. ICDE 2022.
> >
> > [7] Wang et al. Next-item Recommendation with Sequential Hypergraphs. SIGIR 2020.

---

> > > ### Author Response · Authors · 2023-08-18
> > > **Reply to Reviewer WQyi**
> > >
> > > Thank you for engaging in a continued discussion regarding potential issues related to baselines.
> > >
> > > We are in agreement that the inclusion of more competitive baselines serves to bolster our claims and demonstrate the effectiveness of our approach. We are grateful to the reviewer for highlighting a pertinent work, FDSA, which regrettably was omitted from the current manuscript. We firmly believe that incorporating this reference will further enhance the strength and novelty of our ideas. Rest assured, we are committed to adding it in the final version.
> > >
> > > However, we would also like to address the reviewer's concerns regarding our present comparative results. It is important to acknowledge that our comparison already encompasses 13 baselines across five distinct techniques (as listed in Figure 1). This thorough evaluation constitutes a robust contribution to the field of session-based recommendations, even when contrasted with the broader literature on sequence recommendations, as you have pointed out. Additionally, it's worth noting that our evaluation is grounded in data of 100 million E-commerce interactions, a scale unparalleled even in papers from the industry. We emphasize that our innovation lies not in solely challenging sequence models, but rather in augmenting them by addressing potential limitations within graph neural networks. Our guiding principle is to explore alternative methods for constructing spatial structures to facilitate anonymous recommendations.
> > >
> > > Turning to the comparison between SASRec and GRU4Rec, we indeed recognize the observed performance disparities. It is important to highlight that we have adopted the cross-entropy loss as the optimization objective for all methods. While the original GRU implementation and paper suggest the use of TOP1 and BRP loss functions, our experimentation revealed that the cross-entropy loss enhances the stability of GRU and even enables it to outperform transformers in scenarios with shorter session lengths. Nonetheless, we acknowledge that GRU still struggles to match the performance of the transformer architecture, as evident in the discrepancies in Beauty, Books, and Electronics categories.
> > >
> > > We greatly value the reviewer's engagement in discussing and identifying distinctions among various architectures. We want to affirm our commitment to defending our stance and refuting unfounded allegations. Our intention is to foster a constructive exchange of ideas that strengthens the rigor of our work.
> > >
> > > Thank you once again for your thoughtful review, which continues to guide our revisions and improvements.

---

> > > > ### Comment · Reviewer_WQyi · 2023-08-19
> > > > **Thank you for the response**
> > > >
> > > > Thank you for your response. You refer the choice of the cross-entropy loss function as a possible source of contrary performance outcomes. More recent works (dating from 2022) [8-10], which also employ the cross-entropy loss function, consistently find that SASRec outperforms GRU4Rec on datasets with short session lengths. Moreover, results from well-known open-source libraries like RecBole (CE loss) [11] and ReChorus (BPR loss) [12] corroborate this observation.
> > > >
> > > > As I've previously mentioned, the results raise concerns about the accuracy and dependability of the results shown. I indeed feel unconformable about the comment on the alleged "unfounded allegations", something neither of us would wish for. If there exists further evidence on how "cross-entropy loss enhances the stability of GRU and even enables it to outperform transformers in scenarios with shorter session lengths", I would be happy to learn about them.
> > > >
> > > > [8] Xia et al. Efficient On-Device Session-Based Recommendation. TOIS.
> > > >
> > > > [9] Hou et al. CORE: Simple and Effective Session-based Recommendation within Consistent Representation Space. SIGIR 2022.
> > > >
> > > > [10] Yang et al. Multi-Behavior Hypergraph-Enhanced Transformer for Sequential Recommendation. KDD 2022.
> > > >
> > > > [11] https://github.com/RUCAIBox/RecBole-GNN/blob/main/results/sequential/diginetica.md
> > > >
> > > > [12] https://github.com/THUwangcy/ReChorus#models

---

> > > > > ### Author Response · Authors · 2023-08-19
> > > > > **Reply to Reviewer WQyi**
> > > > >
> > > > > Thank you for sharing the results of the latest development and benchmark results. I agree with you that revisiting historical publications and implementations would contribute significantly to the research community. I appreciate your concerns regarding the accuracy and dependability of the results presented in the manuscript. Ensuring that the findings are robust and reliable is indeed crucial, and I understand your perspective on this matter. However, it's worth noting that [8] indicates that the authors use the ranking loss for GRU4Rec, and [9] only mentions that the authors leverage RecBole without the loss information. Additionally, we can observe that the performance of GRU4Rec and SASRec in [10] is very close. We're also interested in exploring the differences further.
> > > > >
> > > > > Regarding the claim about the cross-entropy loss enhancing the stability of GRU and enabling it to outperform transformers in scenarios with shorter session lengths, I acknowledge the need for further evidence to substantiate this assertion. I have gone deeper into the existing literature and our implementation to identify studies that provide a more comprehensive understanding of this phenomenon. There are two other possible reasons stemming from the batchifying operations and prediction mechanism. (1) We group sessions with similar lengths into a bucket/batch for training and prediction, which may enhance the robustness of recurrent neural networks. (2) For S3Rec, we follow the same setting as masked language models (predicting the token at the current index), whereas SASRec functions as a generative language model predicting the token at the next time step. Although I believe this difference would not have a significant impact, this difference should be highlighted and estimated again.
> > > > >
> > > > > Lastly, I appreciate your engagement and feedback on the manuscript. Your insights have prompted me to reevaluate and strengthen the arguments presented in the paper to enahance the overall quality.

---

> > > > > > ### Author Response · Authors · 2023-08-21
> > > > > > **Reply to Reviewer WQyi**
> > > > > >
> > > > > > Dear Reviewer WQyi,
> > > > > >
> > > > > > We hope our comprehensive rebuttal has addressed some of your concerns. With the discussion phase deadline drawing near, we kindly request the opportunity to provide further clarification or address any additional questions you may have. Your consideration is greatly appreciated.
> > > > > >
> > > > > > Many thanks,
> > > > > >
> > > > > > Authors of Submission 125

---

### Official Review · Reviewer_BdTQ · 2023-07-06

**Soundness:** 3 good
**Presentation:** 3 good
**Contribution:** 3 good
**Rating:** 6
**Confidence:** 3

**Summary:**

This paper presents a novel framework for session-based recommendations. The code idea of the method is to extract highly frequent attribute patterns from graphs to augment session sequence encoding. Specifically, it first leverages frequent graph pattern mining for attribute pattern retrieval. Then it applies GAT-based encoders for item representations and use these representations as memory to faciliate session-based encoding. Finally, it converts item-side graph representations into sequences and aggregates them with session sequences for transformer-based model training. Empirical results compared with sequence-based and graph-based baselines show the advantages of the method.

**Strengths:**

The paper is well-written wrt the high-level topic and what the high-level intuitions are.
Good motivation of mining attribute features for sequence encoding enhancement, which is a significant problem to the community.
Experiment is extensive and convincing.  Experimental results on two public datasets and one industrial dataset seem promising compared with sequence-based and graph-based baselines.
Comprehensive analysis in the ablation studies and the appendix.

**Weaknesses:**

Some clarifications can be made more clear. For example, it mentions using gSpan to mine patterns and keep patterns in the twenty types as it shows. However, it is not clear why keeping patterns in these twenty types, like it is a common practice or it can faciliate training.
More powerful sequence-based approaches can be discussed.  As far as I know, P5 [1] and M6-rec [2] are more powerful sequence-based baselines. It would be better to discuss these methods.

[1] Geng, Shijie, et al. "Recommendation as language processing (rlp): A unified pretrain, personalized prompt & predict paradigm (p5)." RecSys 2022.
[2] Cui et al. "M6-Rec: Generative Pretrained Language Models are Open-Ended Recommender Systems." Arxiv 2022.


**Questions:**

Being not familiar with graph-based transformers, I mainly have concerns about the design choice mentioned in its method.
- What is the reason of choosing the GAT-based encoder rather than GCN and GraphSAGE, for learning attribute graph representations?
- What is the insights and design intuition of using attribute representations as memory to augment sequence encoding?
- Why using sequences as the final representations rather than aggregate information using graph representations?


**Limitations:**

Design intuition and technical details can be explained more clearly.
It would be better to provide discussion and comparision of more state-of-the-art baselines.

---

> ### Author Rebuttal · Authors · 2023-08-10
>
> **W1. Clarifications for pattern acquisition and baselines**
>
> A1. Upon the motifs containing three or four nodes, we exclusively consider those featuring a circle or a triangle. These two structural types play a crucial role in diminishing randomness and enhancing robustness. In response to the reviewer's suggestions, we will explore and elaborate on recent literature. However, these references lack specificity regarding session-based recommendations, which, in turn, present more formidable challenges due to truncated behavioral histories and the absence of user profiles.
>
> **Q1. Reasons to choose GAT rather than GCN**
>
> A1. We tested various message-passing methods and found GAT to have higher recall.
>
> **Q2. Insights of memory augmentation**
>
> A2. Relying solely on temporal historical data remains constrained in prior research. Simultaneously, forming graph topologies might introduce noise from random clicks, and using graph neural networks would help counteract over-smoothing. Hence, we suggest incorporating the transformer architecture from a temporal perspective, while combining collaborative details via metadata and frequent attribute patterns in the spatial view.
>
> **Q3. Reasons to choose sequence models**
>
> A3. Recommendations regarding session behaviors must consider temporal signals. Research is ongoing on constructing suitable graph topologies for training and inference. Furthermore, utilizing graph neural networks could help address over-smoothing concerns.

---

> > ### Author Response · Authors · 2023-08-21
> > **Reply to Reviewer BdTQ**
> >
> > Dear Reviewer BdTQ,
> >
> > We hope our comprehensive rebuttal has addressed some of your concerns. With the discussion phase deadline drawing near, we kindly request the opportunity to provide further clarification or address any additional questions you may have. Your consideration is greatly appreciated.
> >
> > Many thanks,
> >
> > Authors of Submission 125

---

> > ### Comment · Reviewer_BdTQ · 2023-08-21
> >
> > Thanks for your efforts. My concerns regarding the design have been addressed and I will keep my rating.

---

### Official Review · Reviewer_2VLV · 2023-07-07

**Soundness:** 3 good
**Presentation:** 3 good
**Contribution:** 2 fair
**Rating:** 4
**Confidence:** 3

**Summary:**

This paper studies session-based recommendation in E-commerce. The authors propose to enhance user intent identification by constructing attribute transition graphs and using frequent attribute patterns as memory to augment session representations. The study employs frequent graph pattern mining algorithms to find consequential graphlets and uses these attribute patterns as accessible memory to augment session sequence encoding. It leverages multi-head graph attention to learn patterns and local session graph representations in the aligned space. The experiments are conducted on two public benchmark datasets and three large-scale industrial datasets, demonstrating notable improvement across various evaluation metrics.

**Strengths:**

1. Session-based recommendation is an important research topic in data mining.
2. The proposed method achieves better results.
3. The authors provide sufficient comparisons with baseline methods.

**Weaknesses:**

1. The motivations for different model components are not clearly stated nor discussed. The model components are directly introduced, without detailed discussions on their intuitions and big pictures. The experimental analysis also does not well elaborate on the reasons and technical findings.

2. The proposed method is complicated and its computational cost is high. Session-based recommendation has strict latency restrictions and it is unclear whether the proposed method meets the requirements. Further efficiency analysis is highly needed, which is important for supporting the main motivation.

3. The evaluation is purely offline and no online experimental result is reported. It is not clear whether the proposed method can be deployed online with acceptable effort.

4. The proposed framework is a combination of mature techniques. Thus, the technical contributions are not quite salient. More empirical analysis could be made.

5. It would be helpful to compare the results on long-tailed/cold-start attributes and items, since the proposed method seems to be suitable for handling these cases.

**Questions:**

How about the computational cost of this work? Is there any analysis of the real training and inference time?

**Limitations:**

The authors mainly discussed the limitations at the technique level, but are not aware of the potential negative societal impact. Some discussions on recommendation fairness and diversity can be added.

---

> ### Author Rebuttal · Authors · 2023-08-10
>
> **W1. Unclear motivation**
>
> A1. In E-commerce services, like scenarios involving new users or those in private mode, session-based recommendation (SBR) is challenging as it doesn't incorporate user profiles. Solely relying on temporal data is a limitation in prior research. Simultaneously, constructing graph topologies might introduce noise due to random clicks; here, utilizing graph neural networks helps address the issue of over-smoothing. In Section 6, our experimental evaluation begins by assessing next-item prediction through metrics like Recall and ranks (MRR and NDCG). We underscore the importance of attribute patterns and our proposed graph-nested attention. Furthermore, we conduct two experiments focused on intent capture by estimating attribute predictions and period-item recommendations. We welcome suggestions and comments aimed at enhancing clarity, if feasible.
>
> **W2. Latency and cost**
>
> A2. Computation encompasses offline and online components. The offline facet involves pattern mining and retrieval, while the online facet entails encoding session data via attribute pattern augmentation. The offline costs can be disregarded, given their nearly linear correlation with data size, not to mention resulting data structures fewer than those in the original sessions. As depicted in Table 6, attributes and patterns number in the thousands, whereas sessions could reach several million. Moreover, graph density of resulting frequent patterns remains stable at around 1.0. In contrast, other methods (e.g., global and shortcut) might exhibit factors of 10x or even 100x. As evident from our runtime, our model operates as efficiently as transformers.
>
> **W3. Online evaluation**
>
> A3. Online assessment of recommendations entails corporate security considerations, necessitating approval from the company.
>
> **W4. Combination of mature techniques**
>
> A4. Regrettably, we differ from the reviewer's perspective. Contemporary data mining technologies find widespread utility across diverse fields. We employ these techniques to reliably extract refined patterns, utilizing them as a repository to enhance recommendations. Furthermore, we introduce graph-nested attention, showcasing its effectiveness by outperforming state-of-the-art approaches by an average of 4.5%. Our framework stands poised to enrich the SBR and graph-related retrieval research community.
>
> **W5. Long-tail and cold-start evaluation**
>
> A5. In Figure 6, we address the cold-start scenario. Short-period cases relate to brief sessions, where our model excels in stability.
>
> **Q1. Computational cost**
>
> A1. Please see W2 and A2 for the cost analysis. Due to the attention mechanism's parallelism, both training and inference can be as efficient as regular transformers. There's an additional memory attention cost, but it's capped at 12 patterns.

---

> > ### Author Response · Authors · 2023-08-21
> > **Reply to Reviewer 2VLV**
> >
> > Dear Reviewer 2VLV,
> >
> > We hope our comprehensive rebuttal has addressed some of your concerns. With the discussion phase deadline drawing near, we kindly request the opportunity to provide further clarification or address any additional questions you may have. Your consideration is greatly appreciated.
> >
> > Many thanks,
> > Authors of Submission 125

---

### Official Review · Reviewer_jZGH · 2023-07-11

**Soundness:** 3 good
**Presentation:** 3 good
**Contribution:** 2 fair
**Rating:** 5
**Confidence:** 1

**Summary:**

The paper proposes a framework that effectively utilizes attribute graph patterns to enhance anonymous sequence encoding for session-based recommendations. Given the lack of personal information in session-based recommendation scenarios, making accurate item suggestions within the session is crucial. The authors performed an extensive experimental evaluation, and it seems there is an improvement. I would appreciate more info on methodology used for experimentation.

Minor comments:

The citations are clickable, which is convenient for further reading.
It would be beneficial to add references to Table 1 for better context.
Providing more explanation in lines 70-75 would enhance clarity and understanding.
While I'm not an expert in the area, it seems that the related work section could benefit from including more recent papers. Currently, there is only one paper from 2023.
Fig 5-6 are clear and easy to read, which is commendable.

**Strengths:**

1. Extensive experimentation



**Weaknesses:**

1. Not clearly demonstrated novelty of the method
2. The paper is a bit hard to follow
3. Some examples of the data can be useful in the paper.

**Questions:**

Can you please list a number of the current practical applications that would benefit from the proposed method?

Could you provide more information about the costs associated with the model? Additionally, I'm interested in knowing about the latency, or response time, of the model.

**Limitations:**

Yes

---

> ### Author Rebuttal · Authors · 2023-08-10
>
> **W1. Unclear novelty**
>
> A1. The session-based recommendation (SBR) does not provide the user profile in E-commerce services, such as for new or private mode users. Solely relying on temporal data remains limited in prior research. Simultaneously, constructing graph topologies might introduce noise from random clicks, while graph neural networks can exacerbate over-smoothing. We thus suggest a transformer architecture for the temporal view, while aggregating collaborative data via meta-data and frequent attribute patterns in the spatial view. Comprehensive experiments underscore the proposal's novelty and effectiveness.
>
> **W2. Hard to follow**
>
> A2. We commence by defining the problem and subsequently introduce the session and transition graph construction procedure. The primary methodology comprises three parts: (1) acquiring frequent attribute patterns through data mining, (2) encoding sessions via relevant pattern retrieval and memory augmentation, (3) providing recommendations through attention. In the experimental section (Section 6), we initially assess next-item prediction by computing Recall and ranks (MRR and NDCG). We demonstrate the significance of attribute patterns and our proposed graph-nested attention. Furthermore, we conduct two experiments on intent capture by estimating attribute predictions and period-item recommendations. We welcome suggestions and comments to enhance clarity, if feasible.
>
> **W3. Example illustration**
>
> A3. In Figure 1 and Figure 2, we present two cases: silver ↔ silver ↔ blue ↔ blue, which illustrates the user's color intent; additionally, the brand pattern Apple ↔ Apple ↔ Samsung suggests a potential change in intent.
>
> **Q1. More practical applications**
>
> A1. Session-based recommendations (SBR) serve as a direct industrial use, while our method aids other graph-related applications, such as answering questions over graphs.
>
> **Q2. Latency and cost**
>
> A2. Computation involves offline and online parts. Offline includes pattern mining and retrieval, while online encodes session data with attribute pattern augmentation. Offline costs are negligible due to linearity with data size, especially compared to resulting structures—far fewer than original sessions. As seen in Table 6, attributes and patterns total several thousand, sessions possibly several million. Graph density for frequent patterns remains stable at around 1.0, in contrast to methods like global or shortcut, which are 10x or 100x. As evident from performance, our model runs as efficiently as transformers.

---

> > ### Comment · Reviewer_jZGH · 2023-08-17
> > **Thank you**
> >
> > I have read the authors' rebuttal, thanks a lot for all the provided explanations. I will change my score accordingly.

---

> > > ### Author Response · Authors · 2023-08-18
> > > **Reply to Reviewer jZGH**
> > >
> > > We extend our gratitude for your alignment with this endeavor and for your active engagement in the analysis. Our confidence remains steadfast in both the strength of our conceptual framework and the validity of our current results. Additionally, the potential of related work within this domain sparks our excitement. Your insightful comments and comprehensive reviews are deeply valued by broadening the reach of our audience and elevating the quality of this paper.

---

### Author Response · Authors · 2023-08-15
**General Comments to Reviewers**

Dear Reviewers,


We sincerely appreciate the time and effort that the reviewers have dedicated to evaluating our paper and raising pertinent questions about its motivation, previous literature, methodology, evaluation, and potential avenues for future research. We are hopeful that our work will undergo a comprehensive and impartial evaluation process. Any lingering concerns or questions that you may have would greatly assist us in enhancing the overall organization and clarity of the manuscript. Consequently, we kindly request your feedback and comments to facilitate a more in-depth discussion.


Thank you for taking the time to review our manuscript!


Best regards,

Authors of Submission 125

---

### Decision · Program_Chairs · 2023-09-21

**Decision:**

Accept (poster)

**Comment:**

Overall somewhat borderline; recommending acceptance mainly due to calibration versus other papers on similar topics, though could go either way. Mostly borderline scores, though one slightly more positive review (though it's the shortest) creates something of a case for acceptance. The rebuttal was very detailed, though not very productive in terms of improving the scores.

Partly recommending acceptance since many of the issues are due to clarifications (though clarity does seem to be something of an issue), which could probably be fixed. Some more substantive issues around motivation/novelty, though these issues were raised by fewer reviewers.